# DP-CSM: Efficient Differentially Private Synthesis for Human Mobility Trajectory with Coresets and Staircase Mechanism

Xin Yao, Juan Yu *, Jianmin Han, Jianfeng Lu, Hao Peng , Yijia Wu and Xiaoqian Cao

College of Mathematics and Computer Science, Zhejiang Normal University, Jinhua 321004, China
* Correspondence: yujuan@zjnu.edu.cn

**Abstract:** Generating differentially private synthetic human mobility trajectories from real trajectories is a commonly used approach for privacy-preserving trajectory publishing. However, existing synthetic trajectory generation methods suffer from the drawbacks of poor scalability and suboptimal privacy–utility trade-off, due to continuous spatial space, high dimentionality of trajectory data and the suboptimal noise addition mechanism. To overcome the drawbacks, we propose DP-CSM, a novel differentially private trajectory generation method using coreset clustering and the staircase mechanism, to generate differentially private synthetic trajectories in two main steps. Firstly, it generates generalized locations for each timestamp, and utilizes coreset-based clustering to improve scalability. Secondly, it reconstructs synthetic trajectories with the generalized locations, and uses the staircase mechanism to avoid the over-perturbation of noises and maintain utility of synthetic trajectories. We choose three state-of-the-art clustering-based generation methods as the comparative baselines, and conduct comprehensive experiments on three real-world datasets to evaluate the performance of DP-CSM. Experimental results show that DP-CSM achieves better privacy–utility trade-off than the three baselines, and significantly outperforms the three baselines in terms of efficiency.

**Keywords:** differential privacy; trajectory publication; coresets; staircase mechanism; trajectory privacy

## 1. Introduction

With the prevalence of positioning-enabled mobile devices, such as smart phones and watches, huge amounts of moving trajectories of individuals in the geospatial space have been collected. These trajectories are valuable for various applications, such as intelligent transportation [1] and urban planning [2,3]. However, as trajectories contain sensitive information, directly releasing or sharing them might pose serious privacy threats to individuals. Adversaries can identify individuals, and further infer their home addresses, health status and hobbies [4–6] from their trajectories. Therefore, it is crucial to designing effective privacy protection methods to sanitize trajectories before publishing or sharing.

Many efforts have been devoted to privacy preserving trajectory data publishing. Early work adopts partition-based privacy models [7,8], such as *k*-anonymity [9], *l*-diversity [10] and *t*-closeness [11], to sanitize trajectories. However, the above-mentioned privacy models suffer drawbacks of depending on background knowledge of adversaries, being vulnerable to various attacks [12,13] and insufficient theoretical privacy guarantee. Consequently, differential privacy emerged as a strict privacy model independent of background knowledge, and is widely applied in privacy-preserving trajectory data publishing [14–20].

However, directly applying differential privacy on trajectory data is challenging. Due to the intrinsic high dimensionality and continuous infinite spatial (location) space of trajectory data, it requires extensive noise to ensure differential privacy, resulting in poor utility of released synthetic trajectory. To overcome the challenges, existing research concentrates on generating synthetic trajectories with high utility under the differentially private constraint. From the perspective of spatial space discretization, existing differentially private

trajectory generation methods can be categorized into two classes, i.e., grid-partition-based and location-generalization-based methods.

Grid-partition-based methods could be further divided into two subclasses, i.e., prefix-tree-based and Markov-model-based methods. The prefix tree is the most commonly used index structure in differentially private trajectory sanitizing mechanisms for answering various counting queries and generating synthetic trajectories. Chen et al. [15] first introduced the noisy prefix tree to sanitize trajectory data. To improve efficiency and data utility, He et al. [17] proposed the DPT framework, which adopts a hierarchical reference system to discrete GPS trajectory data, and utilizes a direction-weighted sampling mechanism to further improve the data utility of synthetic trajectories. SafePath [21] also models trajectories in a prefix tree. Recently, Cai et al. [22] proposed a new method to construct the noisy prefix tree to save privacy budget. The above mechanisms compactly encode the sequential and count information of the raw trajectories into a noisy prefix tree, and then reconstruct and release sanitized trajectories according to the tree. It is expected that the sanitized trajectories preserve high utility for count queries and frequent pattern mining tasks. However, the node count decreases drastically with the growth of the prefix tree. Consequently, the added noise is relatively large compared to the count, resulting in high utility loss of the released trajectory data.

To remedy the deficiencies of the grid-partition-based methods and improve the utility of synthetic trajectories, several Markov-model-based methods have been proposed. Gursory et al. [23] proposed the DP-Star (differentially private synthetic trajectory publisher) framework. DP-Star discretizes location space with adaptive grids, and generates synthetic trajectories using differentially private statistics, including trip distribution, route length distribution, state transition matrix. Ghane et al. [19] designed a graphical generative model, TGM, to generate differentially private synthetic trajectories with stay information preserved.

The above-mentioned methods discretize or compress location space with data-independent grid partition, which makes it hard to preserve the spatial density distribution of trajectories. To overcome this shortcoming, several location-generalization-based methods are proposed. Hua et al. [24] proposed the first differentially private generalization algorithm for releasing trajectories, which clustered trajectories close to each other using the exponential mechanism. To overcome the shortcomings of [24], Li et al. [25] proposed a bounded Laplace noise-based differentially private trajectory data publishing mechanism to improve the utility of reconstructed trajectories. The advantage of the generalization-based method lies in its data-dependent density-aware location universe compression. However, the differentially private clustering subprocedure of the above methods is computationally expensive.

Although there have been previous works on differentially private trajectory data publishing, these works suffer from suboptimal data utility and weak scalability. Motivated by high efficiency of coresets in clustering big data [26] and the lower noise cost of the staircase mechanism [27], we propose a novel generalization-based differentially private trajectory publishing mechanism using coreset-based clustering and the staircase noise-adding mechanism, named DP-CSM, to provide high scalability and utility while guaranteeing $\epsilon$-differential privacy. The DP-CSM consists of two stages: location generalization and trajectory reconstruction. In the location generalization stage, we construct a coreset for locations of original trajectories at each timestamp. Compared with the existing location generalization methods, the coresets can greatly improve the efficiency of $k$-means clustering [28,29]. In the stage of trajectory reconstruction, the staircase mechanism is chosen as the noise-adding mechanism instead of the Laplace mechanism. This is because the staircase mechanism adds less noise than the Laplace mechanism under the same privacy budget [27], thereby maintaining high data utility.

In particular, the contributions of this paper are as follows.

- We introduce coreset-based clustering into the location generalization stage to improve the efficiency and scalability of synthetic trajectory generation. Coresets, instead of the

original dataset, are used for clustering, as they achieve similar utility and the same privacy level with the original dataset.

- We utilize the staircase mechanism, instead of the traditional Laplace mechanism, to perturb the counts of trajectories in the trajectory reconstruction stage to avoid adding excessive noise, thereby preserving high data utility under the same privacy budget and achieving better privacy and utility trade-off than existing methods.
- We provide theoretical proof that the proposed DP-CSM satisfies $\epsilon$-differential privacy. Since trajectory reconstruction satisfies differential privacy, the DP-CSM also satisfies differential privacy (See Section 4.4 for detailed analysis).
- We conduct comprehensive experiments on three real-world trajectory datasets to evaluate the performance of the proposed method in terms of data utility and efficiency.

## 2. Related Work

Many efforts have been devoted to differentially private data publishing. In this section, we summarize two kinds of closely related work—differentially private trajectory data publishing and differentially private sequential data publishing.

### 2.1. Differentially Private Trajectory Data Publishing

Existing work on differentially private trajectory data publishing can be divided into three categories according to the materials used for generating trajectories.

#### 2.1.1. Noisy Prefix Tree

Noisy prefix tree is the most commonly used data structure to capture sequential and counts information of trajectories. Chen [15] et al. proposed the first trajectory publishing mechanism with differential privacy. They make use of the noisy prefix tree to put the trajectory sequence with the same prefix into the same branch, so as to reduce the output field and ensure the high time efficiency of the trajectory publishing process. They use two sets of inherent constraints of prefix trees for constraint inferences, which improves the data utility. However, the number of sequences that belong to a same branch decreases quickly as the prefix tree grows. Thus, the data utility will be reduced. Chen et al. [16] then used the variable-length $n$-gram model to extract the basic information of the trajectory database and process the general trajectory data. They used a Markov-based search tree to reduce noise and improve the utility of the data. Recently, there have been some works on synthesis trajectory publishing. DPT [17] discretizes the trajectories at multiple resolutions using hierarchical reference systems to capture individual movements at differing speeds, and constructs one prefix tree for each resolution.

#### 2.1.2. Private Statistics

To overcome the deficiency of the noisy prefix tree-based methods, there are recent works which generally generate synthetic trajectories using private statistics extracted from the original trajectories. Typical examples are the AdaTrace [18], DP-star [23] and TGM [19]. DP-Star uses the minimum description length metric to summarize raw trajectories using their representative points, thereby achieving a desirable trade-off between the preciseness and conciseness of their information content. AdaTrace is a scalable location trajectory synthesizer consists of three novel features: deterministic attack, provable statistical privacy and strong utility preservation. AdaTrace generates a generative model through four steps: feature extraction, synopsis learning, utility and privacy preserving noise injection, and generation of synthetic location trajectory with differential privacy. The output trajectories preserve critical utility information in original trajectories, and are robust against existing trajectory attacks. DP-Star is a framework for differentially private trajectory publication. DP-Star relies on several components such as Minimum Description Length metric, density-aware grid, trip distribution and median length estimation. TGM first encodes the data as a graphical generative model and privately generates synthetic trajectories, which achieves substantially high computational efficiency and utility.

### 2.1.3. Cluster Centers

To take full advantage of bias distribution of locations in trajectories for efficient location universe compression, Hua et al. [24] proposed a generalization-based approach for differentially private trajectory publishing. It first generalizes the trajectories by merging the locations at the same time, and then releases trajectories after generalization in a differentially private manner. To improve the efficiency of private location cluster and data utility, Li et al. [25] proposed a bounded Laplace noise-based trajectory data publication mechanism with differential privacy. The design of this noise generation mechanism enables the noise added to the real trajectory count to be sampled in a legal range to enhance differential privacy. In addition, they remove the partition strategy that removes the original trajectory, which improves the utility of data and efficiency of publication. Generalization-based approaches are scalable to large trajectory datasets.

### 2.2. Differentially Private Sequential Data Publishing

Research on differentially private sequential data publishing is currently considered in two settings: centralized setting and multi-party setting. Most works focused on the centralized setting. Chen et al. [30] proposed an algorithm that releases a prefix tree of sequences to support count queries and frequent string mining. Chen et al. [16] proposed a variable-length n-gram model to balance the privacy–utility trade-off. However, the privacy budget consumption of the above method closely depends on the height of the noisy prefix tree. To disentangle the privacy budget from the height of the constructed tree, Zhang et al. [31] proposed the PrivTree model, which used only a constant amount of noise in constructing the prediction suffix tree. Most recently, Tang et al. [32] explored differentially private multi-party sequential data publishing, and proposed a distributed prediction suffix tree to solve the problem. However, the above approaches for sequential data publishing cannot be directly applied on trajectory data due to their ignorance of temporal information and spatial–temporal correlation.

## 3. Preliminaries

In this section, we introduce some basic definitions.

**Definition 1** (Trajectory [24]). *A trajectory of an individual is a chronologically ordered sequence of time–location pairs, which can be formally represented as $T = (t_1, l_1) \rightarrow (t_2, l_2) \rightarrow \cdots \rightarrow (t_{|T|}, l_{|T|})$, where $t_i$ is the i-th sampled timestamp, $l_i$ is the location (represented by the latitude and longitude coordinates) of the individual at $t_i$, where $1 \leq i \leq |T|$, and $|T|$ is the length of the trajectory.*

A trajectory dataset $\mathcal{D}$ refers to a set of trajectories collected from different individuals. For simplicity, we assume that all trajectories in the dataset $\mathcal{D}$ are sampled at the same sequence of timestamps, i.e., trajectories are synchronously recorded and have the same length. The sampled timestamps of each trajectory in $\mathcal{D}$ could be represented as $Time(\mathcal{D}) = \{t_1, t_2, \cdots, t_{|T|}\}$, where $|T|$ is the number of sampled timestamps. Locations at each timestamp $t_i \in Time(\mathcal{D})$ can be represented as a location set $\mathcal{D}^i$.

**Definition 2** ($\epsilon$-differential privacy [33]). *Given a randomized algorithm Ag, $\mathcal{O}_{Ag}$ is the set of all possible outputs of the Ag. For any two adjacent datasets $\mathcal{D}$ and $\mathcal{D}'$ (differing on at most one record) and any subset $O \subseteq \mathcal{O}_{Ag}$, if and only if the algorithm Ag satisfies*

$$Pr[Ag(\mathcal{D}) \in O] \leq \exp(\epsilon) \times Pr[Ag(\mathcal{D}') \in O], \tag{1}$$

*then the algorithm Ag provides $\epsilon$-differential privacy, where $\epsilon$ is the privacy budget, and a smaller $\epsilon$ indicates a higher privacy level.*

Differential privacy is guaranteed by an optimal noise-adding mechanism. The Laplace mechanism, the exponential mechanism and the staircase mechanism are the commonly adopted noise-adding mechanisms in existing works. In this paper, we choose the staircase mechanism as the noise-adding mechanism, as it retains higher data utility than others.

**Definition 3** (Query sensitivity [27]). *Given a query function $f : \mathcal{D} \rightarrow \mathbb{R}^d$ which returns $\mathcal{D}$-dimensional numerical vectors as outputs, the sensitivity of $f$ is defined as*

$$\Delta f = \max_{\mathcal{D}, \mathcal{D}'} ||f(\mathcal{D}) - f(\mathcal{D}')||_1, \tag{2}$$

*where $\mathcal{D}$ and $\mathcal{D}'$ are neighboring datasets, and $|| \cdot ||_1$ is the $\ell_1$-norm.*

**Definition 4** (Staircase mechanism [27]). *Given a query function $f : \mathcal{D} \rightarrow \mathbb{R}^d$, its sensitivity $\Delta f$ and the privacy budget $\epsilon$, the staircase mechanism is to add noise to the query result, i.e.,*

$$K(\mathcal{D}) = f(\mathcal{D}) + g(\Delta f, \epsilon), \tag{3}$$

*where $g(\Delta f, \epsilon)$ is the noise generator sampling noise from an $\mathcal{D}$-dimensional staircase-shaped probability distribution.*

The probability density function of the staircase-shaped probability distribution is defined as follows:

$$h_\gamma(X; \Delta f, \epsilon) = \begin{cases} e^{-j\epsilon}\alpha(\gamma) & ||X||_1 \in [j\Delta f, (j+\gamma)\Delta f] \\ e^{-(j+1)\epsilon}\alpha(\gamma) & ||X||_1 \in [(j+\gamma)\Delta f, (j+1)\Delta f] \end{cases}, \tag{4}$$

where $j \in \mathbb{N}$, and $\gamma \in [0, 1]$. $\alpha(\gamma)$ is the normalization factor to make $\int \int \cdots \int_{\mathbb{R}^d} h_\gamma(x) dx_1 dx_2 \ldots dx_d = 1$,

$$\alpha(\gamma) = \frac{d!}{2^d (\Delta f)^d \sum_{j=1}^d \frac{d!}{j!(d-j)!} c_{d-j}(b + (1-b)\gamma^j)}, \tag{5}$$

where $b = e^{-\epsilon}$, $c_j = \sum_{i=0}^{+\infty} i^j b^j$ and $\gamma = \frac{1}{1+e^{\epsilon/2}}$.

## 4. DP-CSM

### 4.1. The Framework

Given a trajectory dataset $\mathcal{D}$, our goal is to sanitize it into a differentially private trajectory dataset $\tilde{\mathcal{D}}$ while retaining its data utility as high as possible. To achieve the goal, we propose a new method with the combination of a coreset-based location generalization and staircase mechanism called DP-CSM. The architecture of the DP-CSM is shown in Figure 1, and it consists of two stages. The first stage is timestamp-wise location generalization, which inputs the location datasets of all timestamps and outputs their corresponding generalized location datasets. The second stage is to reconstruct trajectories according to the resulting sets of generalized locations while guaranteeing that the reconstruction process satisfies differential privacy.

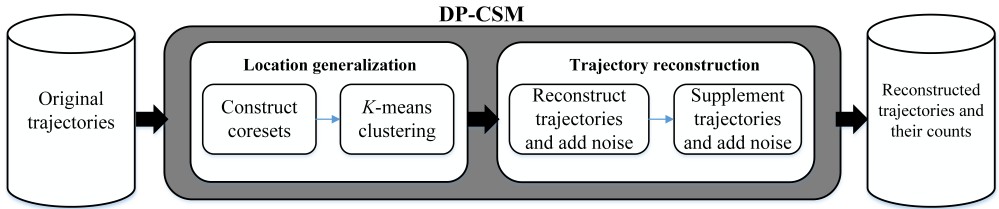

**Figure 1.** Framework of the DP-CSM. (The location generalization module utilizes coresets-based clustering to improve efficiency. The trajectory reconstruction module reconstructs synthetic trajectories with the generalized locations and uses the staircase mechanism to realize differential privacy.)

### 4.2. Coreset-Based Location Generalization

As a result of the quadratic complexity of the existing clustering-based location generalization algorithms [20,24,25], location generalization for each timestamp is a bottleneck of the existing methods. To improve the efficiency, we propose the coreset-based algorithm. Coresets are small, weighted summaries of a large dataset such that solutions found on the coresets are competitive with solutions found on the full dataset [26]. We formally describe the coreset-based location generalization algorithm in Algorithm 1. In the following, we will elaborate on the coreset-based location generalization algorithm.

---

**Algorithm 1:** coresets based location generalization algorithm

---

**Input** : $\mathcal{D}, k, m$

**Output:** $\mathcal{L} = \{L^1, \ldots, L^{|Time(\mathcal{D})|}\}$

1 **for** *location dataset $\mathcal{D}^i$ of each timestamp in $\mathcal{D}$* **do**
2     **for** *each location $l$ in $\mathcal{D}^i$* **do**
3         compute $s(l)$; // Compute sensitivity of $l$ according to [28,29]
4     **end**
5     **for** *each $l \in \mathcal{D}^i$* **do**
6         $p(l) \leftarrow s(l) / \sum_{l' \in \mathcal{D}^i} s(l')$; // Nomalization
7     **end**
8     $C^i \leftarrow$ Sample $m$ weighted points from $\mathcal{D}^i$ where each point $x$ has weight $\frac{1}{m \cdot p(l)}$ and is sampled with probability $p(l)$;
9     $L^i \leftarrow$ Perform $k$-means on the coresets $C^i$ to obtain $k$ cluster centers as generalized locations;
10 **end**
11 **return** $\mathcal{L}$;

---

For Algorithm 1, the input parameters are the original trajectory dataset $\mathcal{D}$, the number of cluster centers $k$, and the size of coresets $m$. For each location set $\mathcal{D}^i$ of timestamp $t_i$ in $\mathcal{D}$, the coreset-based location generalization algorithm efficiently outputs $k$ cluster centers as generalized locations. Steps 2–6 of Algorithm 1 represent the coreset construction procedure following the algorithm proposed in [28,29]. Specifically, we first compute the sensitivity $s(l)$ of each point $l \in \mathcal{D}^i$ (the sensitivity $s(l)$ mentioned here is different from the sensitivity in differential privacy). After computing the $s(l)$, we need to perform importance sampling and each point $l$ is sampled with probability $p(l)$. This process is repeated until $C^i$ consists of $m$ points and each sampled point has a weight $\frac{1}{m \cdot p(l)}$. Then, we perform $k$-means on the coresets $C^i$ to obtain $k$ centers and output. Finally, we repeat the above process $|T|$-1 times.

To illustrate the rationale of Algorithm 1, we give a toy example. Suppose that we have a location set containing 7 locations, as shown in Figure 2. In Figure 2, we can see that Algorithm 1 samples $m$ locations to represent the original set. For example, we sample the location $l_{12}$ to represent locations $l_{11}, l_{12}, l_{13}, l_{14}$, thus, the weight of $l_{12}$ after sampling is 4 (because we use one location to represent four locations). Finally, we add the location $l_{12}$ and its weight 4 into the coreset. The other situations are shown in Table 1.

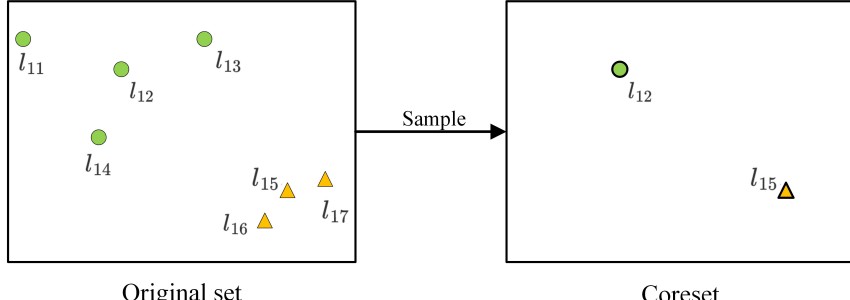

**Figure 2.** Construction of coresets. An original set of locations (circles and triangles on the left rectangle) and its coreset (bold circle and triangles on the right rectangle) are shown in Figure 2. The elements $l_{12}$ and $l_{15}$ in the coreset are the corresponding representations of the original set: $l_{12}$ represents the locations $l_{11}, l_{12}, l_{13}, l_{14}$, and $l_{15}$ represents the locations $l_{15}, l_{16}, l_{17}$.

**Table 1.** Construction of coresets in Figure 2.

| Locations in Coresets | Locations in Original Set | Weight before Sampling | Weight after Sampling |
|---|---|---|---|
| $l_{12}$ | $l_{11}, l_{12}, l_{13}, l_{14}$ | 1 | 4 |
| $l_{15}$ | $l_{15}, l_{16}, l_{17}$ | 1 | 3 |

### 4.3. Trajectory Reconstruction

The trajectory reconstruction stage is to differential-privately reconstruct trajectories from the resulting generalized locations of Algorithm 1. We formally describe the procedure in Algorithm 2. The algorithm takes the original trajectory dataset $\mathcal{D}$, generalized location sets $\mathcal{L}$ and the privacy budget $\epsilon$ as inputs, and outputs the final reconstructed trajectories with the $\epsilon$-differential privacy guarantee. Specifically, the reconstruction algorithm consists of three major steps, i.e., candidate trajectory generation, differentially private trajectory selection, and trajectory supplementation.

---

**Algorithm 2:** Trajectory reconstruction algorithm

**Input** : $\mathcal{D}, \mathcal{L} = \{L^1, \ldots, L^{|Time(\mathcal{D})|}\}, \epsilon$.
**Output:** $\widetilde{\mathcal{D}}$: reconstructed trajectories and their noisy counts.

1　$\widetilde{\mathcal{D}} \leftarrow \emptyset$; // Initialize an empty trajectory set
2　$\mathcal{T} = L^1 \times L^2 \times \cdots \times L^{|Time(\mathcal{D})|}$; //Construct all possible candidate trajectories
3　**for** *each candidate reconstructed trajectory $\widetilde{T}$ in $\mathcal{T}$* **do**
4　　　$\mathcal{O}_{\widetilde{T}} \leftarrow Query(\widetilde{T}, \mathcal{D})$;
5　　　**if** $|\mathcal{O}_{\widetilde{T}}| > 0$ **then**
6　　　　　$n_{\widetilde{T}} = |\mathcal{O}_{\widetilde{T}}| + g(\Delta f, \epsilon)$; // add staircase noise to the counts
7　　　　　$\widetilde{\mathcal{D}} = \widetilde{\mathcal{D}} \cup \{\widetilde{T} \times n_{\widetilde{T}}\}$; // add $n_{\widetilde{T}}$ reconstructed trajectories $\widetilde{T}$ into $\widetilde{\mathcal{D}}$
8　　　**end**
9　**end**
10　**if** $|\widetilde{\mathcal{D}}| < |\mathcal{D}|$ **then**
11　　　//Supplement trajectories
12　　　randomly sample $|\mathcal{D}| - |\widetilde{\mathcal{D}}|$ trajectories from $\mathcal{T} - \widetilde{\mathcal{D}}$;
13　**end**
14　**return** $\widetilde{\mathcal{D}}$; // Output reconstructed trajectories and their noisy counts

---

Candidate trajectories are generated according to the generalized location sets via the set Cartesian product alike operation, as shown in step 2 of Algorithm 2, i.e., $\mathcal{T} = L^1 \times L^2 \times \cdots \times L^{|Time(\mathcal{D})|}$. Then, we choose reconstructed trajectories from $\mathcal{T}$ in a differentially private manner, as shown in Steps 3–9 of Algorithm 2. If a candidate trajectory covers at least one

original trajectory in the trajectory dataset $\mathcal{D}$, then it is selected as a reconstructed trajectory. By covering an original trajectory we means that the location of each timestamp is covered in the generalization location. Specifically, given an candidate trajectory $\widetilde{T} = \{\widetilde{l_1}, \widetilde{l_2}, \ldots, \widetilde{l_{|T|}}\}$, and an original trajectory $T = \{l_1, l_2, \ldots, l_{|T|}\}$, we call $\widetilde{T}$ covers $T$, if $\forall i \in \{1, 2, \ldots, |T|\}$, $l_i \in \widetilde{l_i}$. To facilitate description, we define $Query(T, \mathcal{D})$ as a query function to search covered trajectories for the reconstructed trajectory $\widetilde{T}$ from $\mathcal{D}$. To guarantee differential privacy, we add a staircase noise to the true number of covered trajectories, and construct the corresponding amount of reconstructed trajectories. In particular, the noisy count is calculated as follows.

$$n_{\widetilde{T}} = |Query(\widetilde{T}, \mathcal{D})| + g(\Delta f, \epsilon), \tag{6}$$

where $n_{\widetilde{T}}$ is the noisy count of the reconstructed trajectory.

As the noise addition during the reconstructed trajectory selection procedure, the number of reconstructed trajectories would be less than that of the original trajectory dataset. Thus, Steps 10–13 of the algorithm are designed to generate supplement trajectories to make the reconstructed trajectory dataset have the same number of trajectories as the original dataset.

In particular, suppose we use $\widetilde{\mathcal{D}} = \{\widetilde{T_1}, \widetilde{T_2}, \cdots, \widetilde{T_j}\}$ to represent the reconstructed trajectories passed by the original trajectories, and it is sorted according to its noisy count, that is, $Cnt_1 > Cnt_2 > \cdots > Cnt_j$, where $Cnt_i$ represents the noisy count of the reconstructed trajectory $\widetilde{T_i}$. Next, starting from the interval $(Cnt_2, Cnt_1]$, we calculate the $Num_i$ of the trajectory in the set $\mathcal{T} - \widetilde{\mathcal{D}}$ with the noisy count in the interval $(Cnt_{i+1}, Cnt_i]$, the calculation method is as follows:

$$Num_i = |\mathcal{T} - \widetilde{\mathcal{D}}| \cdot \int_{Cnt_{i+1}}^{Cnt_i} h_\gamma(X; \Delta f, \epsilon) dx. \tag{7}$$

where $\Delta f$ is the global sensitivity, $\epsilon$ is the privacy budget, and $h_\gamma(X; \Delta f, \epsilon)$ represents the probability density function of the staircase mechanism. The count of reconstructed trajectory in the set $\mathcal{T} - \widetilde{\mathcal{D}}$ is 0 (because no original trajectory has passed through). After adding noise that satisfies the staircase mechanism, the probability that the trajectory in $\mathcal{T} - \widetilde{\mathcal{D}}$ is counted in the interval $(Cnt_{i+1}, Cnt_i]$ is $\int_{Cnt_{i+1}}^{Cnt_i} h_\gamma(X; \Delta f, \epsilon) dx$.

To reconstruct sufficient trajectories, we finally randomly sample $Num_i$ trajectories from $\mathcal{T} - \widetilde{\mathcal{D}}$, and add those trajectories to the final published trajectory dataset. Their noisy counts are random values in the corresponding interval $(Cnt_{i+1}, Cnt_i]$. When the total count of the trajectory dataset to be published reaches the total count of the original dataset, Algorithm 2 stops.

In the following, we will illustrate the procedure of Algorithm 2 by giving a simple example. We consider a trajectory dataset containing eight trajectories, as shown in Figure 3a. From Figure 3b, we can see that our proposed location generalization algorithm based on coresets partitions the original location set at each timestamp into two clusters, and each cluster is replaced by a cluster center. Then, Algorithm 2 connects the $k$ centers at different timestamps in chronological order to reconstruct the trajectory and output its noisy count. For example, we connect the centers $l_{11}, l_{21}, l_{31}$ to obtain the reconstructed trajectory $l_{11} \to l_{21} \to l_{31}$. The reconstructed trajectory $l_{11} \to l_{21} \to l_{31}$ has been passed through by $T_1, T_2$, thus, the true count of $l_{11} \to l_{21} \to l_{31}$ is 2. Then, we add noise satisfying the staircase mechanism to the true count 2. Finally, we output the reconstructed trajectory $l_{11} \to l_{21} \to l_{31}$ and its noisy count 1. The other situations are shown in Table 2.

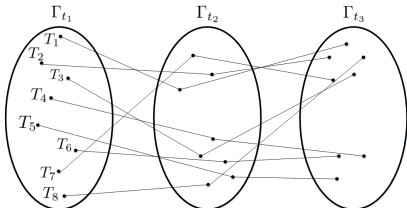
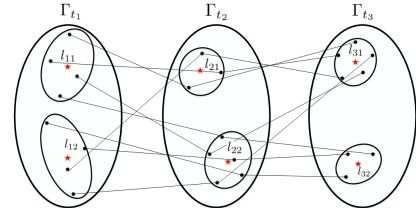

| (**a**) An example dataset with 8 trajectories. | (**b**) Location Generalization. |

**Figure 3.** An illustration of the trajectory reconstruction algorithm.

**Table 2.** Differentially private publication of the sample data in Figure 3.

| Generalized Trajectories | Raw Trajectories | Real Counts | Noisy Counts |
|---|---|---|---|
| $l_{11} \to l_{21} \to l_{31}$ | $T_1, T_2$ | 2 | 1 |
| $l_{11} \to l_{21} \to l_{32}$ | *Null* | 0 | 2 |
| $l_{11} \to l_{22} \to l_{31}$ | $T_3$ | 1 | 0 |
| $l_{11} \to l_{22} \to l_{32}$ | $T_4$ | 1 | 1 |
| $l_{12} \to l_{21} \to l_{31}$ | $T_7$ | 1 | 0 |
| $l_{12} \to l_{21} \to l_{32}$ | *Null* | 0 | 0 |
| $l_{12} \to l_{22} \to l_{31}$ | $T_8$ | 1 | 3 |
| $l_{12} \to l_{22} \to l_{32}$ | $T_5, T_6$ | 2 | 1 |

*4.4. Privacy Analysis*

We now analyze the privacy guarantee of DP-CSM from a theoretical perspective. DP-CSM consists of two main stages. The first stage is a coreset-based *k*-means clustering, which can be regarded as a no-privacy algorithm $Ag_1$. The second algorithm $Ag_2$ utilizes the staircase mechanism to add noise to the counts of trajectories. We first prove the following theorem.

**Theorem 1.** *Given a trajectory dataset $\mathcal{D}$ and its generalized location sets $\mathcal{L}$, let $NC_{\mathcal{T}}(\mathcal{D}, \mathcal{L})$ denote the output of the Algorithm 2, and $\mathcal{P}_{NC_{\mathcal{T}}}$ be the set of all possible outputs of the $NC_{\mathcal{T}}(\mathcal{D}, \mathcal{L})$. For any two adjacent datasets $\mathcal{D}$ and $\mathcal{D}'$ (differing on at most one record), two generalized location sets $\mathcal{L}$ and $\mathcal{L}'$ ($\mathcal{L}$ generated from $\mathcal{D}$ and $\mathcal{L}'$ generated from $\mathcal{D}'$), any output $r \subseteq \mathcal{P}_{NC_{\mathcal{T}}}$, the Algorithm 2 satisfies $\epsilon$-differential privacy if and only if:*

$$Pr[NC_{\mathcal{T}}(\mathcal{D}, \mathcal{L}) = r] \leq Pr[NC_{\mathcal{T}}(\mathcal{D}', \mathcal{L}') = r] \cdot e^{\epsilon}. \tag{8}$$

**Proof of Theorem 1.** Assuming that dataset $\mathcal{D}$ and dataset $\mathcal{D}'$ are adjacent datasets, that is, $\mathcal{D}$ and $\mathcal{D}'$ have only one different trajectory $T_x$, we use $\widetilde{T}_x$ to represent the generalized trajectory of trajectory $T_x$, and $NC_{\mathcal{T}}^i(\mathcal{D}, \mathcal{L})$ represents the noisy count of generalized trajectory $\widetilde{T}_i \in \mathcal{T}$, then the probability that the noisy count $NC_{\mathcal{T}}^i(\mathcal{D}, \mathcal{L})$ equals $r = \{cnt_1, cnt_2, \cdots, cnt_{|\mathcal{D}|}\}$ is

$$
\begin{aligned}
Pr[NC_{\mathcal{T}}(\mathcal{D}, \mathcal{L}) = r] &= Pr[NC_{\mathcal{T}}(\mathcal{D}, \mathcal{L}) = \{cnt_1, cnt_2, \cdots, cnt_{|\mathcal{D}|}\}] \\
&= Pr[NC_{\mathcal{T}}^1(\mathcal{D}, \mathcal{L}) = cnt_1] \times Pr[NC_{\mathcal{T}}^2(\mathcal{D}, \mathcal{L}) = cnt_2] \times \cdots \times \\
&\quad Pr[NC_{\mathcal{T}}^{|\mathcal{D}|}(\mathcal{D}, \mathcal{L}) = cnt_{|\mathcal{D}|}] \\
&= \prod_{i=1}^{|\mathcal{D}|} Pr[NC_{\mathcal{T}}^i(\mathcal{D}, \mathcal{L}) = cnt_i].
\end{aligned} \tag{9}
$$

We discuss the following three cases.

**Case 1:** For any generalized trajectory $\widetilde{T}_i \neq \widetilde{T}_x$, we can derive that

$$Pr[NC_{\mathcal{T}}^i(\mathcal{D}, \mathcal{L}) = cnt_i] = Pr[NC_{\mathcal{T}}^i(\mathcal{D}', \mathcal{L}') = cnt_i], \tag{10}$$

**Case 2:** For any generalized trajectory $\widetilde{T}_i = \widetilde{T}_x \wedge \widetilde{T}_x \in \widetilde{\mathcal{D}}$, the count of the generalized trajectory $\widetilde{T}_x$ is obtained by adding noise that satisfies the staircase mechanism on the basis of the real count. According to staircase mechanism [27], we can derive that

$$Pr[NC_{\mathcal{T}}^x(\mathcal{D}, \mathcal{L}) = cnt_x] \leq Pr[NC_{\mathcal{T}}^x(\mathcal{D}', \mathcal{L}') = cnt_x] \cdot e^{\epsilon}, \tag{11}$$

**Case 3:** For an arbitrary generalized trajectory $\widetilde{T}_i = \widetilde{T}_x \wedge \widetilde{T}_x \notin \widetilde{\mathcal{D}}$, it can be divided into two sub-cases $cnt_x \in (Cnt_{min}, Cnt_1)$ and $cnt_x \notin (Cnt_{min}, Cnt_1)$.

**(a)** $cnt_x \in (Cnt_{min}, Cnt_1)$

Assuming $cnt_x \in (Cnt_{i+1}, Cnt_i)$, from the analysis of Section 4.3 and the probability density function of the staircase mechanism, we can derive that:

$$
\begin{aligned}
Pr[NC_{\mathcal{T}}^x(\mathcal{D}', \mathcal{L}') = cnt_x] &= \frac{1}{Cnt_i - Cnt_{i+1}} \cdot \int_{Cnt_{i+1}-1}^{Cnt_i-1} h_{\gamma}(X; \Delta f, \epsilon) dx \\
&= \frac{1}{Cnt_i - Cnt_{i+1}} \cdot [(j+y)\Delta f - (Cnt_{i+1} - 1)] \cdot a(\gamma)e^{-j\epsilon} + \\
&\quad [(Cnt_i - 1) - (j+y)\Delta f] \cdot a(\gamma)e^{-(j+1)\epsilon} \\
&= \frac{1}{Cnt_i - Cnt_{i+1}} \cdot [(j+y)\Delta f - Cnt_{i+1} + 1] \cdot a(\gamma)e^{-j\epsilon} + \\
&\quad [Cnt_i - 1 - (j+y)\Delta f] \cdot a(\gamma)e^{-(j+1)\epsilon} \\
&= \frac{1}{Cnt_i - Cnt_{i+1}} \cdot [(j+y)\Delta f - Cnt_{i+1}] \cdot a(\gamma)e^{-j\epsilon} + \\
&\quad [Cnt_i - (j+y)\Delta f] \cdot a(\gamma)e^{-(j+1)\epsilon} + \\
&\quad a(\gamma)e^{-j\epsilon} - a(\gamma)e^{-(j+1)\epsilon} \\
&\geq \frac{1}{Cnt_i - Cnt_{i+1}} \cdot [(j+y)\Delta f - Cnt_{i+1}] \cdot a(\gamma)e^{-j\epsilon} + \\
&\quad [Cnt_i - (j+y)\Delta f] \cdot a(\gamma)e^{-(j+1)\epsilon} + \\
&\quad a(\gamma)e^{-j\epsilon}e^{-\epsilon} - a(\gamma)e^{-(j+1)\epsilon} \\
&= \frac{1}{Cnt_i - Cnt_{i+1}} \cdot [(j+y)\Delta f - Cnt_{i+1}] \cdot a(\gamma)e^{-j\epsilon} + \\
&\quad [Cnt_i - (j+y)\Delta f] \cdot a(\gamma)e^{-(j+1)\epsilon} \\
&= Pr[NC_{\mathcal{T}}^x(\mathcal{D}, \mathcal{L}) = cnt_x].
\end{aligned}
\tag{12}
$$

That is:

$$Pr[NC_{\mathcal{T}}^x(\mathcal{D}', \mathcal{L}') = cnt_x] \geq Pr[NC_{\mathcal{T}}^x(\mathcal{D}, \mathcal{L}) = cnt_x], \tag{13}$$

where $h_{\gamma}(X; \Delta f, \epsilon)$ represents the probability density function of the staircase mechanism.

**(b)** $cnt_x \notin (Cnt_{min}, Cnt_1)$

We use $Cnt_{min}$ to represent the minimum noisy count of the trajectory in the output trajectory dataset, then:

$$
\begin{aligned}
Pr[NC_{\mathcal{T}}^x(\mathcal{D}', \mathcal{L}') = cnt_x] &= 1 - \int_{Cnt_{min}-1}^{Cnt_1-1} h_{\gamma}(X; \Delta f, \epsilon) dx \\
&= 1 - \{[(j+y)\Delta f - (Cnt_{min} - 1)]a(\gamma)e^{-j\epsilon} + \\
&\quad [(Cnt_1 - 1) - (j+y)\Delta f]a(\gamma)e^{-(j+1)\epsilon}\} \\
&= 1 - \{[(j+y)\Delta f - Cnt_{min} + 1]a(\gamma)e^{-j\epsilon} + \\
&\quad [Cnt_1 - 1 - (j+y)\Delta f]a(\gamma)e^{-(j+1)\epsilon}\} \\
&= 1 - [(j+y)\Delta f - Cnt_{min}]a(\gamma)e^{-j\epsilon} - \\
&\quad [Cnt_1 - (j+y)\Delta f]a(\gamma)e^{-(j+1)\epsilon} - \\
&\quad a(\gamma)e^{-j\epsilon} + a(\gamma)e^{-(j+1)\epsilon} \\
&\leq 1 - [(j+y)\Delta f - Cnt_{min}]a(\gamma)e^{-j\epsilon} - \\
&\quad [Cnt_1 - (j+y)\Delta f]a(\gamma)e^{-(j+1)\epsilon} - a(\gamma)e^{-j\epsilon}e^{-\epsilon} + \\
&\quad a(\gamma)e^{-(j+1)\epsilon} \\
&= 1 - [(j+y)\Delta f - Cnt_{min}]a(\gamma)e^{-j\epsilon} - \\
&\quad [Cnt_1 - (j+y)\Delta f]a(\gamma)e^{-(j+1)\epsilon} \\
&= Pr[NC_{\mathcal{T}}^x(\mathcal{D}, \mathcal{L}) = \varnothing].
\end{aligned}
\tag{14}
$$

That is:

$$Pr[NC_{\mathcal{T}}^x(\mathcal{D}', \mathcal{L}') = cnt_x] \leq Pr[NC_{\mathcal{T}}^x(\mathcal{D}, \mathcal{L}) = cnt_x]. \tag{15}$$

Thus we can derive that $Pr[NC_{\mathcal{T}}^x(\mathcal{D}', \mathcal{L}') = cnt_x] = Pr[NC_{\mathcal{T}}^x(\mathcal{D}, \mathcal{L}) = cnt_x]$.

Combining the three cases, we can derive $Pr[NC_{\mathcal{T}}(\mathcal{D}, \mathcal{L}) = r] \leq Pr[NC_{\mathcal{T}}(\mathcal{D}', \mathcal{L}') = r] \cdot e^\epsilon$, thus, Algorithm 2 satisfies $\epsilon$-differential privacy according to Definition 2. □

**Theorem 2.** *We denote DP-CSM as $Ag$ and assume $P_{Ag}$ is the set of all possible outputs of the $Ag$. For any two adjacent datasets $\mathcal{D}$ and $\mathcal{D}'$ and any output $O \subseteq P_{Ag}$, the DP-CSM satisfies $\epsilon$-differential privacy if and only if:*

$$Pr[Ag(\mathcal{D}) = O] \leq Pr[Ag(\mathcal{D}') = O] \cdot e^\epsilon. \tag{16}$$

**Proof of Theorem 2.** Assuming that $\mathcal{D}$ and $\mathcal{D}'$ are two adjacent datasets, we use $Ag_1$ to represent our proposed location-generalized algorithm and $O_1$ to represent its output; We use $Ag_2$ to represent the trajectory reconstruction algorithm, and $O_2$ to represent its output. Ag represents the entire model, and $O$ represents its output. We can derive that:

$$Pr[Ag(\mathcal{D}) = O] = Pr[Ag_1(\mathcal{D}) = O_1] \cdot Pr[Ag_2(\mathcal{D}) = O_2]. \tag{17}$$

According to Definition 2 and Theorem 1, we have:

$$\begin{aligned} Pr[Ag_1(\mathcal{D}) = O_1] \cdot Pr[Ag_2(\mathcal{D}) = O_2] \ &\leq Pr[Ag_1(\mathcal{D}') = O_1] \cdot (Pr[Ag_2(\mathcal{D}') = O_2] \cdot e^\epsilon) \\ &= Pr[Ag(\mathcal{D}') = O] \cdot e^\epsilon. \end{aligned}$$

$$\tag{18}$$

Combining Equations (17) and (18), we have:

$$Pr[Ag(\mathcal{D}) = O] \leq Pr[Ag(\mathcal{D}') = O] \cdot e^\epsilon. \tag{19}$$

Finally, we prove that the DP-CSM satisfies $\epsilon$-differential privacy according to the definition of $\epsilon$-differential privacy. □

*4.5. Complexity Analysis*

The computational cost of DP-CSM mainly relates to the location generalization and reconstruction of trajectories.

Firstly, we analyze the computational complexity of the location generalization, which is mainly composed of coreset construction and $k$-means clustering. The coreset construction is performed at each timestamp, and its computational complexity is $O(k|\mathcal{D}||T|)$, where $|\mathcal{D}|$ is the number of trajectories, $|T|$ is the length of trajectory, $k$ is the number of clusters. The complexity of performing $k$-means is $O(nkm|T|)$, where $n$ is the number of iterations of $k$-means. The time complexity of Algorithm 1 is $O(nkm|T|)$. Additionally, the time complexity of traditional $k$-means is $O(nk|\mathcal{D}||T|)$, which means Algorithm 1 is faster than traditional $k$-means ($m \leq |\mathcal{D}|$).

Secondly, we connect the center of each cluster at each timestamp. Since the number of centers is $|\mathcal{D}||T|$, the computational complexity of connecting the centers is $O(|\mathcal{D}||T|)$. The complexity of adding noise and supplementing trajectories is less than $|\mathcal{D}|$, thus, the computational complexity of trajectory reconstruction algorithm is $O(|\mathcal{D}||T|)$.

Therefore, the total computational complexity of DP-CSM is $O(nkm|T|)$.

**5. Experiment**

In this section, we empirically evaluate the performance of our method DP-CSM in terms of the data utility of sanitized trajectory datasets and the scalablity for coping with large trajectory datasets.

*5.1. Experiment Setup*

We compare DP-CSM with three representative works (INFOCOM15 [24], IS17 [25] and PCG [20]). We implemented all methods in Python, and all experiments were run on a computer with an Intel Core i7-9700 CPU, 16G RAM.

5.1.1. Datasets

We used three publicly available real-life trajectory databases: T-drive (https://www.microsoft.com/en-us/research/publication/t-drive-trajectory-data-sample/, accessed on 5 December 2022) [34,35], Geolife https://www.microsoft.com/en-us/download/details.aspx?id=52367, accessed on 5 December 2022) [36–38] and Roma (http://crawdad.org/roma/taxi/20140717/taxicabs/index.html, accessed on 5 December 2022) [39], in our experiments.

**T-Drive.** A dataset contains the GPS trajectories of 10,357 taxis in Beijing from 2 February to 8 February. We chose the trajectory from 8:30 to 14:30 as our experimental data, and each trajectory contains 32 locations. The time interval between any two adjacent locations in a trajectory is 10 min. After preprocessing, we finally selected 12,000 trajectories as our experimental data.

**Geolife.** A dataset was collected through different GPS loggers and GPS phones in the Geolife project by 182 users in a period of over five years (from April 2007 to August 2012). The Geolife trajectory dataset contains 17,621 trajectories with a total distance of 1,292,951 km and a total duration of 50,176 h. After preprocessing, we finally selected 12,000 trajectories as our experimental data, and each trajectory contains 32 locations. The time interval between any two adjacent locations in one trajectory is 10 s.

**Roma taxi.** A dataset contains mobility traces of taxi cabs in Rome, Italy. It contains GPS trajectories of 320 taxis collected over 30 days (from 1 February 2014 to 2 March 2014). After preprocessing, we finally select 12,000 trajectories as our experimental data, and each trajectory contains 32 locations. The time interval between any two adjacent locations in one trajectory is approximately 15 s.

5.1.2. Data Utility Metrics

In the experiments, we utilize three commonly-used metrics (i.e., spatial distribution similarity, Hausdorff distance and range query distortion), and three entropies introduced in [40] to comprehensively evaluate the data utility preservation capability of our proposed method from different aspects.

**Spatial distribution similarity.** The spatial distribution similarity between the raw trajectory database and the sanitized trajectory database is demonstrated with heatmaps. We divide the spatial space into $40 \times 40$ grids and count the number of locations falling in each grid cell. As each heatmap can be represented as an 1600-dimensional vector $(x_1, x_2, \cdots, x_{1600})$, where $x_i$ is the number of locations in the $i$-th grid cell, we further quantify the similarity with the cosine similarities between heatmaps of raw trajectory database and its corresponding sanitized trajectory databases. A sanitized trajectory database which has higher spatial distribution similarity with the raw trajectory database retains higher data utility.

**Hausdorff distance.** Hausdorff distance is a commonly used metric to measure data utility of sanitized trajectory database. The definition of Hausdorff distance is:

$$H(|\mathcal{D}|, |\tilde{\mathcal{D}}|) = \max(h(|\mathcal{D}|, |\tilde{\mathcal{D}}|), h(|\tilde{\mathcal{D}}|, |\mathcal{D}|)), \tag{20}$$

where $h(|\tilde{\mathcal{D}}|, |\mathcal{D}|) = \max_{T \in |\tilde{\mathcal{D}}|} \{\min_{T' \in |\mathcal{D}|} \{Distance(T, T')\}\}$. $Distance(T, T')$ is calculated by the sum of the Euclidean distance of each timestamp between $T$ and $T'$. It reflects the trajectory distance between the original dataset $\mathcal{D}$ and the published dataset $|\tilde{\mathcal{D}}|$, and lower distance implies higher data utility.

**Range query distortion (*RQ*).** Range query distortion is another commonly used metric of the utility of published trajectory data [20,24,41]. It is calculated as follows

$$RQ = \frac{|Q(\mathcal{D}) - Q(\widetilde{\mathcal{D}})|}{max(|Q(\mathcal{D})|, |Q(\widetilde{\mathcal{D}})|)}, \tag{21}$$

where $Q(\mathcal{D})$ is the query result on the raw trajectories, $Q(\widetilde{\mathcal{D}})$ is the query result on the reconstructed trajectories, $Q(\mathcal{D}) - Q(\widetilde{\mathcal{D}})$ is the set difference which returns locations in $Q(\mathcal{D})$ while not in $Q(\widetilde{\mathcal{D}})$, and $|\cdot|$ returns the cardinality of a set. Larger-range query distortion reflects lower data utility.

**The random entropy.** The random entropy can capture the predictability degree of the user's whereabouts if each location is visited with the same probability [42]. The definition of the random entropy is:

$$S_i^{rand} \equiv \log_2 N_i, \tag{22}$$

where $N_i$ is the number of locations which are visited by user $i$.

**The temporal-uncorrelated entropy.** The temporal-uncorrelated entropy can characterize the heterogeneity of visitation patterns [42]. The definition of the temporal-uncorrelated entropy is:

$$S_i^{unc} \equiv - \sum_{j=1}^{N_i} p_i(j) \log_2 p_i(j), \tag{23}$$

where $p_i(j)$ is the historical probability that location $j$ was visited by the user $i$.

**The actual entropy.** The actual entropy depends not only on the frequency of visitation, but also on the order in which the nodes were visited and the time cost at each location, thus capturing the full spatio-temporal order present in a person's mobility pattern [42]. The definition of the actual entropy is:

$$S_i \equiv - \sum_{T_i' \subset T_i} P(T_i') \log_2 [P(T_i')], \tag{24}$$

where $P(T_i')$ is the probability of finding a particular time-ordered subsequence $T_i'$ in the trajectory $T_i$.

### 5.2. Comparison of Data Utility

In this subsection, we first evaluate the utility of the published trajectories with cosine similarity, Hausdorff distance and range query distortion. Then, we evaluate the utility with three entropies introduced in [40]. Finally, we evaluate the trade-off between utility and privacy.

#### 5.2.1. Spatial Distribution Similarity

Figure 4 shows the heatmaps of three real-world trajectory datasets and sanitized trajectory datasets generated by the four models. For each heatmap of a sanitized trajectory dataset, we also calculate its cosine similarity with the corresponding heatmap of raw trajectory dataset. Higher cosine similarity implies better spatial distribution preservation. From Figure 4, we can see that, sanitized trajectory datasets resulting from the proposed DP-CSM have higher cosine similarity scores than those of INFOCOM15 and IS17. Although the cosine similarity scores of DP-CSM are lower than those of the PCG, their gaps are smaller than those of INFOCOM15 and IS17. The reason for the similarity degradation of the proposed DP-CSM is that it adopts location coresets for clustering instead of the original location sets, and coresets are a kind of loss compression of original location sets. This implies that the DP-CSM sacrifices acceptable data utility for the efficiency. The reason for DP-CSM outperforming INFOCOM15 and IS17 is that DP-CSM utilize the staircase

mechanism instead of the Laplacian mechanism, and the staircase mechanism can maintain higher utility by avoiding adding excessive noise to the counts of trajectories.

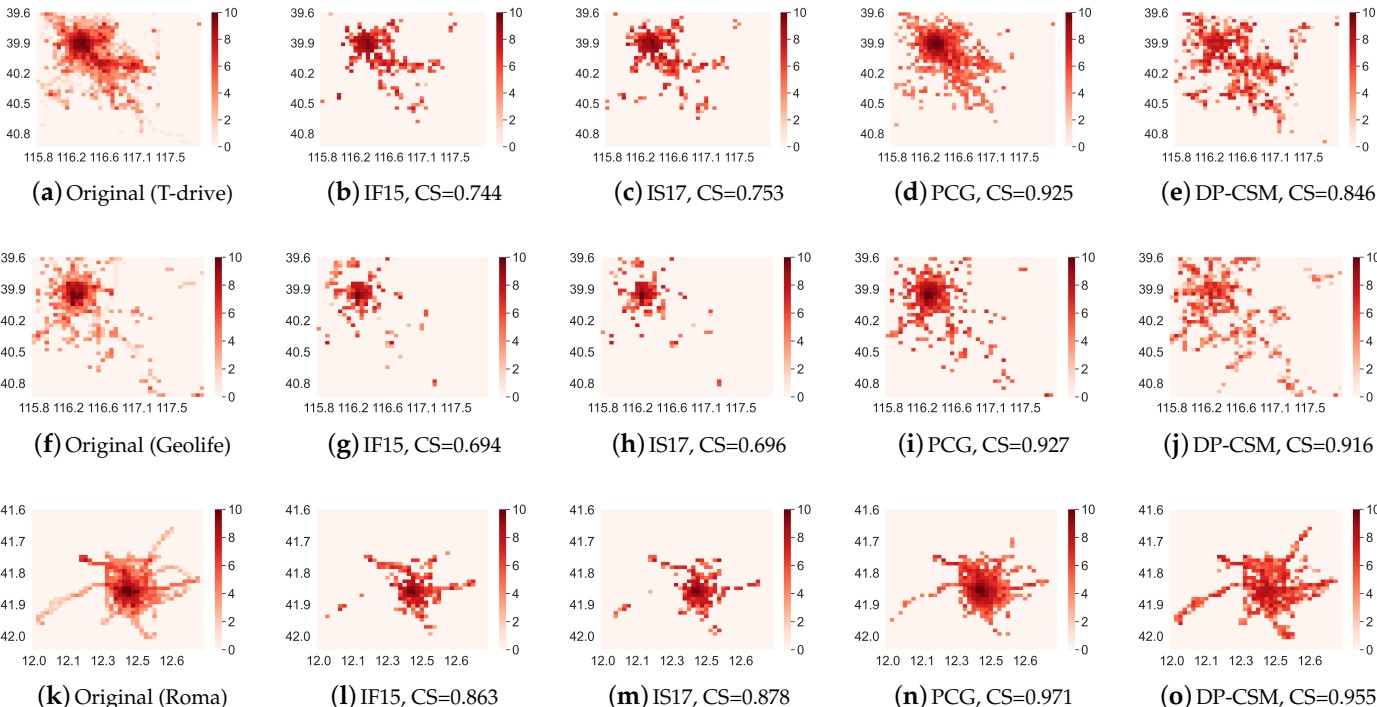

**Figure 4.** Heatmap of raw trajectory databases and sanitized trajectory datasets resulting from the four methods. All the sanitized trajectory datasets were generated under the following parameters setting: $\epsilon = 0.4$, $k = 80$, $|\mathcal{D}| = 12,000$. (**a**–**e**) are results on T-drive, (**f**–**j**) are results on Geolife and (**k**–**o**) are results on Roma. For each figure, the x-axis represents the longitude, the y-axis represents the latitude. The CS in captions of sub-figures stands for the cosine similarity between the heatmap per se and its corresponding raw dataset's heatmap.

### 5.2.2. Hausdorff Distance

Figure 5 shows Hausdorff distance comparisons between the four models under different settings. Hausdorff distance measures the difference between the raw trajectory dataset and a sanitized trajectory dataset, and a smaller distance implies better utility preservation of a sanitized trajectory dataset. From Figure 5, we can see that our DP-CSM has smaller distances than INFOCOM15 and IS17 models have in most cases, which implies better data utility than that of INFOCOM15 and IS17 models. Similar to the trend shown in Figure 4, DP-CSM has larger distances than PCG. This is reasonable since (1) the counts of trajectories have no impact on Hausdorff distance, which means the staircase noise on counts will be useless; (2) the construction of coresets distorts the reconstructed trajectory datasets.

### 5.2.3. Range Query Distortion

Figure 6 presents the comparison results of the four models on range query distortion. Smaller range query distortion implies better utility preservation of a sanitized trajectory dataset. As shown in Figure 6, our DP-CSM has smaller range query distortion than the three baselines do under the same settings. These results demonstrate that although our DP-CSM sacrifices data utility for efficiency, it would not distort the raw dataset too much. In addition, we can also see that the range query distortion decreases with the increasing of the size of datasets. This means that larger trajectory datasets are beneficial for privacy preservation.

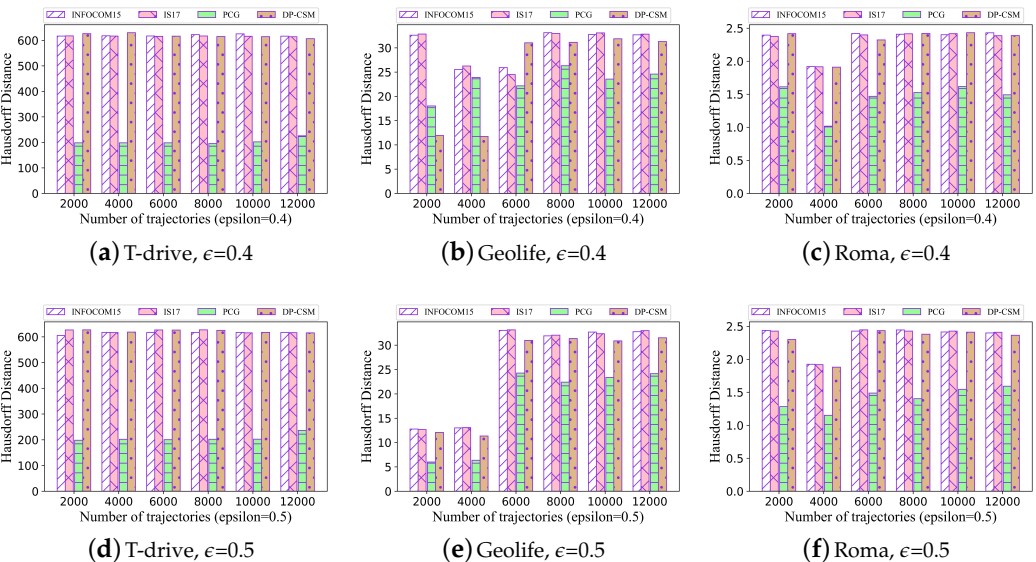

**Figure 5.** Hausdorff distance comparison of the four models under different settings. In particular, we fix $k = 80$ and vary the privacy budget $\epsilon$ and the size of trajectory datasets. For the privacy budget, we choose two kinds of values, i.e., $\epsilon = 0.4$ and $\epsilon = 0.5$, respectively. Under each privacy budget, we vary the sizes of trajectory datasets.

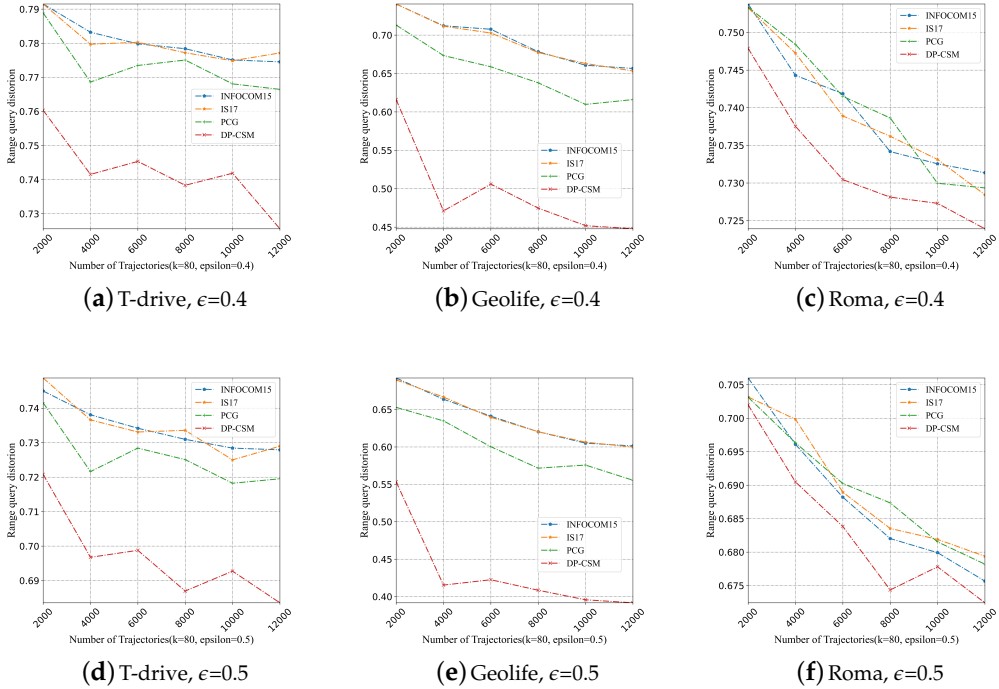

**Figure 6.** Range query distortion comparison of the four models under different settings. In particular, we fix $k = 80$ and vary the privacy budget $\epsilon$ and the size of trajectory datasets. For the privacy budget, we choose two kinds of values, i.e., $\epsilon = 0.4$ and $\epsilon = 0.5$, respectively. Under each privacy budget, we vary the sizes of trajectory datasets.

### 5.2.4. Random Entropy

Figure 7 shows the random entropy comparison results of the four methods with that of the raw trajectory dataset. From Figure 7, we can see that DP-CSM, INFOCOM15 and IS17 could achieve similar random entropy which is larger than the one PCG achieves on the T-drive dataset. Whereas, on the Geolife and Roma datasets, the DP-CSM demonstrates

higher random entropy than others. (The effect on predictability is different on different datasets due to different sampling time intervals. The predictability of trajectories with smaller sampling time intervals may be more susceptible.) This means that the predictability preservation capability of DP-CSM falls behind that of the other methods. It it worth noting that, as the three datasets used in our experiments have finer-grained GPS trajectories compared with the CDR (Call Detailed Record) trajectories used in [42], their predictability might be lower than that of CDR trajectories.

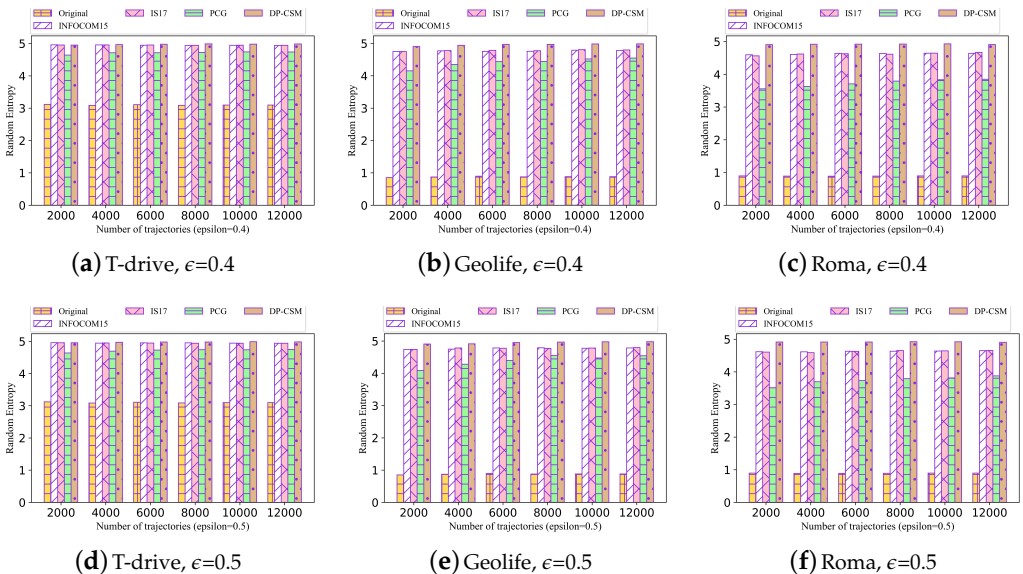

**Figure 7.** Random entropy comparison of the four models under different settings. In particular, we fix $k = 80$ and vary the privacy budget $\epsilon$ and the size of trajectory datasets. For the privacy budget, we choose two kinds of values, i.e., $\epsilon = 0.4$ and $\epsilon = 0.5$, respectively. Under each privacy budget, we vary the sizes of trajectory datasets.

### 5.2.5. Temporal-Uncorrelated Entropy

Figure 8 shows the temporal-uncorrelated entropy comparison results of the four methods with that of the raw trajectory dataset. From Figure 8, we can see similar trends as that present in Figure 7. It also demonstrates that DP-CSM has higher temporal-uncorrelated entropy than other methods. Thus, DP-CSM loses more predictability than the other methods from the data utility perspective, but it could preserve higher privacy as predictability might cause privacy disclosure in some situations.

### 5.2.6. Actual Entropy

Figure 9 demonstrates the actual entropy comparison results of the four methods with that of the raw trajectory dataset. As shown in Figure 9, the four methods have similar actual entropy under different settings, which means they have a similar predictability preservation capability. In addition, gaps between the raw dataset's entropy and sanitized datasets' entropies on Geolife and Roma datasets are larger than those of T-drive. The reason might be the difference of sampling intervals, and the T-drive dataset has a larger sampling interval than the Geolife and Roma datasets.

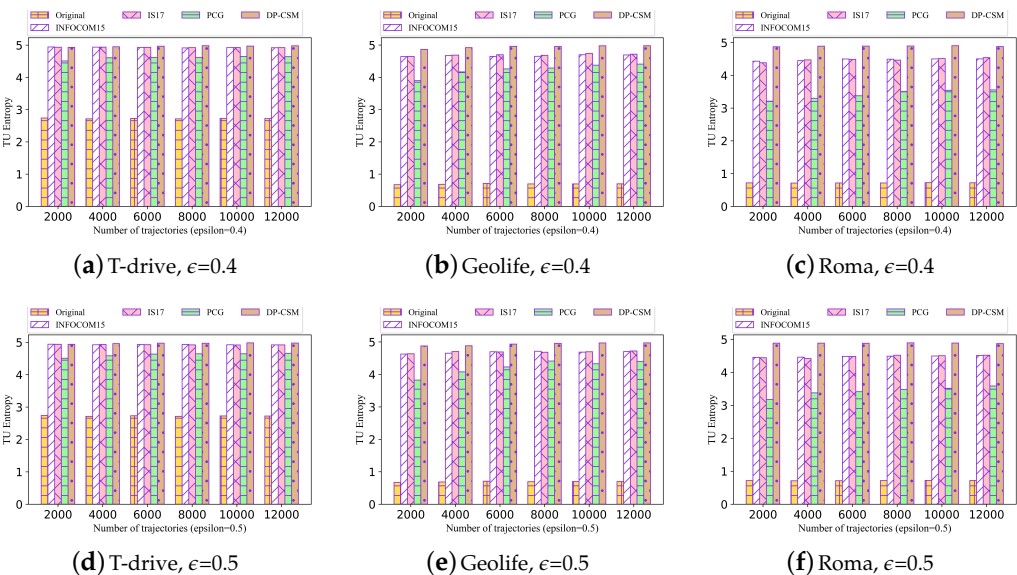

**(a)** T-drive, $\epsilon$=0.4      **(b)** Geolife, $\epsilon$=0.4      **(c)** Roma, $\epsilon$=0.4

**(d)** T-drive, $\epsilon$=0.5      **(e)** Geolife, $\epsilon$=0.5      **(f)** Roma, $\epsilon$=0.5

**Figure 8.** The temporal-uncorrelated entropy comparison of the four models under different settings. In particular, we fix $k$ = 80, and varies privacy budget $\epsilon$ and the size of trajectory datasets. For the privacy budget, we choose two kinds of values, i.e., $\epsilon = 0.4$ and $\epsilon = 0.5$, respectively. Under each privacy budget, we vary the sizes of trajectory datasets.

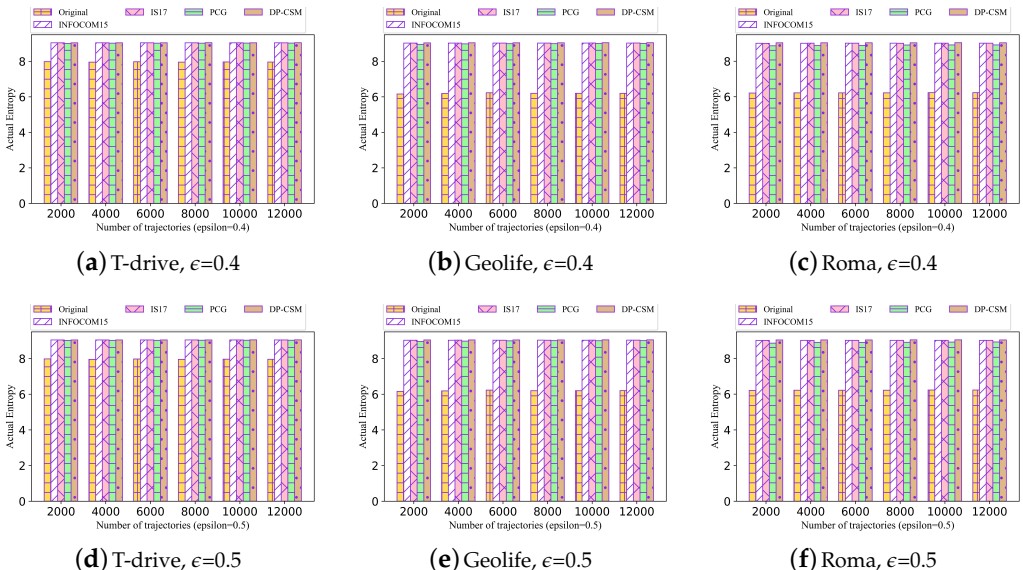

**(a)** T-drive, $\epsilon$=0.4      **(b)** Geolife, $\epsilon$=0.4      **(c)** Roma, $\epsilon$=0.4

**(d)** T-drive, $\epsilon$=0.5      **(e)** Geolife, $\epsilon$=0.5      **(f)** Roma, $\epsilon$=0.5

**Figure 9.** The actual entropy comparison of the four models under different settings. In particular, we fix $k$ = 80 and vary the privacy budget $\epsilon$ and the size of trajectory datasets. For the privacy budget, we choose two kinds of values, i.e., $\epsilon = 0.4$ and $\epsilon = 0.5$, respectively. Under each privacy budget, we vary the sizes of trajectory datasets.

### 5.2.7. Impacts of $\epsilon$ on Data Utility

Figure 10a–c shows the data utility of various method over three real-life trajectory datasets. Figure 10a–c shows results from T-drive, Geolife and Roma, respectively. Figure 10a–c shows that the Hausdorff distance of DP-CSM is larger than PCG's but close to INFOCOM15's and IS17's, which means there is more data utility loss than for PCG but similar utility loss to that for INFOCOM15, IS17. This is reasonable since (1) the counts of trajectories have no impact on Hausdorff distance, which means the staircase noise on counts will be useless; (2) the construction of coresets will distort the trajectory data.

As shown in Figure 10d–f, the horizontal axis represents the privacy budget $\epsilon$, and the vertical axis represents the range query distortion. Figure 10d–f shows that the utility of reconstructed trajectories increases with the $\epsilon$ and DP-CSM has higher utility than the other three works in most cases. This is reasonable because (1) the privacy budget increases means that less noises were added, and therefore utility improves; (2) coreset-based $k$-means converges with fewer iterations. Thus, when the maximum number of iterations is fixed, coreset-based $k$-means have higher utility than $k$-means; (3) compared with the Laplace mechanism, the staircase mechanism can improve the utility by avoiding adding excessive noise to the counts of trajectories.

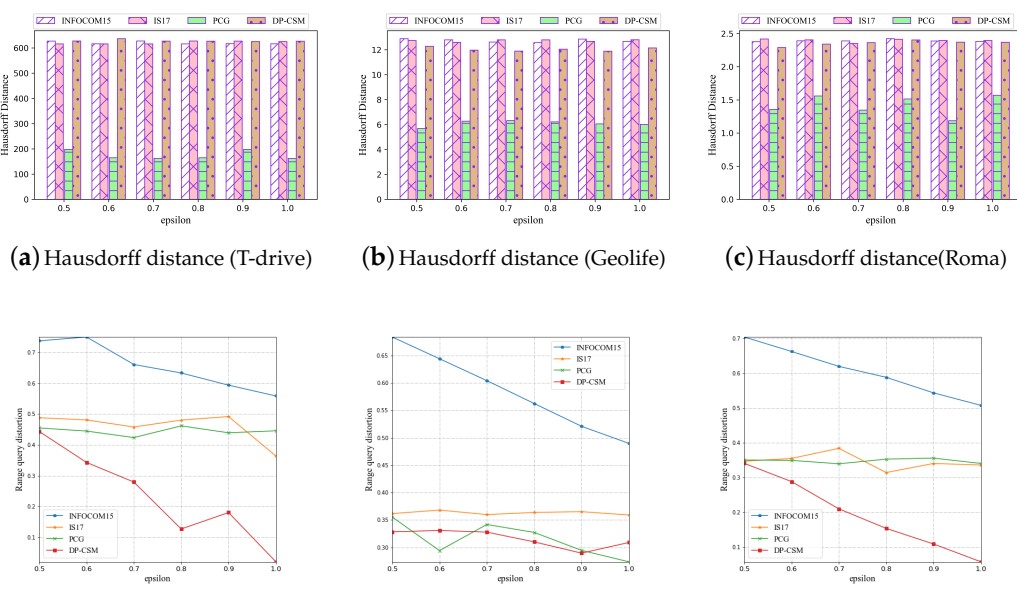

(**a**) Hausdorff distance (T-drive)  (**b**) Hausdorff distance (Geolife)  (**c**) Hausdorff distance(Roma)

(**d**) Range query distortion (T-drive)  (**e**) Range query distortion(Geolife)  (**f**) Range query distortion(Roma)

**Figure 10.** Impacts of privacy budget $\epsilon$ on data utility. We use the Hausdorff distance and range query distortion to quantify data utility of sanitized trajectory datasets resulting from the four methods. In these experiments, we fix $k = 80$, and $|\mathcal{D}| = 2000$ for each dataset. We vary the privacy budget $\epsilon$ from 0.5 to 1.0, and compare the variations of the two data utility metrics for the four methods.

*5.3. Comparison of Scalability*

We study the runtime under different dataset sizes and different number of cluster centers. For the more comprehensive analysis of DP-CSM, we analyze the total trajectory generation time and compare the time efficiency of DP-CSM with INFOCOM15, IS17 and PCG.

5.3.1. Impact of $|D|$ on Scalability

Figure 11 shows the efficiency comparison between the four methods over three real-life trajectory datasets. The time complexity of INFOCOM15, IS17 and PCG is $O(nk|\mathcal{D}||T|)$, and the time complexity of DP-CSM is $O(nkm|T|)$, where $m = 0.2|\mathcal{D}|$. As shown in Figure 11, the DP-CSM is always much faster than INFOCOM15, IS17 and PCG under the same settings. These results demonstrate that the efficiency of DP-CSM is the highest among compared works. This is reasonable since coreset is a small summary of original dataset (thus $m < |\mathcal{D}|$), which reduces the time of performing coreset-based $k$-means.

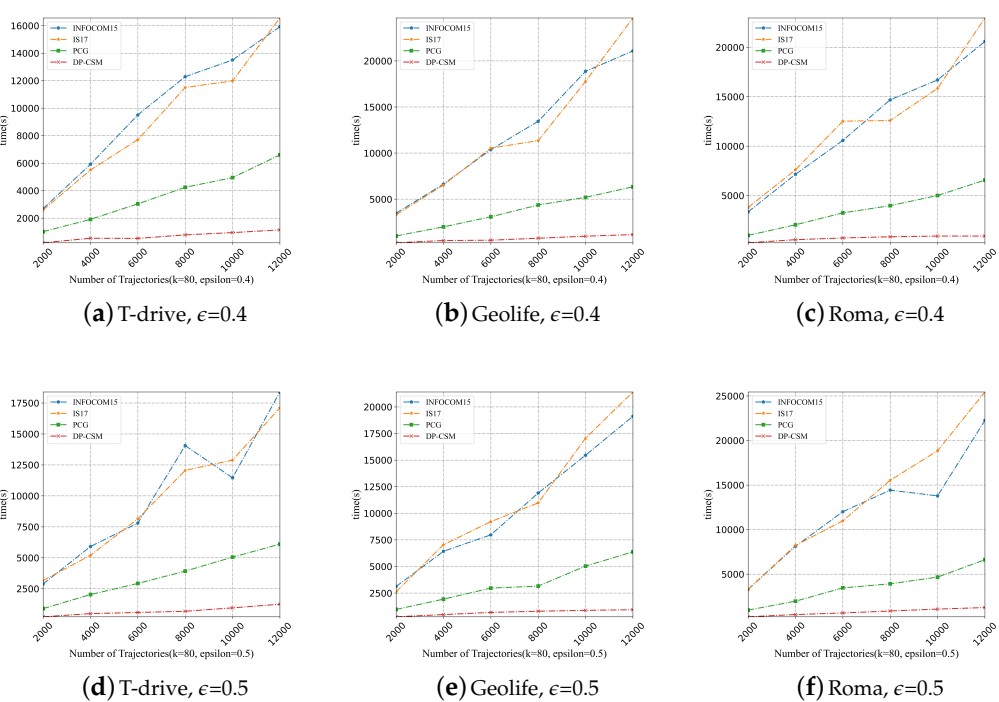

**Figure 11.** Impact of dataset size on the efficiency of the four methods. In these experiments, we fix $k = 80$, and choose two kinds of privacy budget, i.e., $\epsilon = 0.4$, and $\epsilon = 0.5$. For each privacy budget and dataset, we vary the size $|D|$ from 2000 to 12000, and compare the running times of the four methods.

### 5.3.2. Impact of $k$ on Scalability

Figure 12 shows the efficiency of the four methods over three real-life trajectory datasets. As shown in Figure 12, the running time increases approximately linearly with the increasing of $k$. This is reasonable because the time of $k$-means increases linearly as $k$ increases, and the total running time is mainly composed of the time of $k$-means. In addition, we can also see that the DP-CSM is always much faster than INFOCOM15, IS17 and PCG, which means the efficiency of DP-CSM is highest among compared works. The reason is the same as in Section 5.3.1.

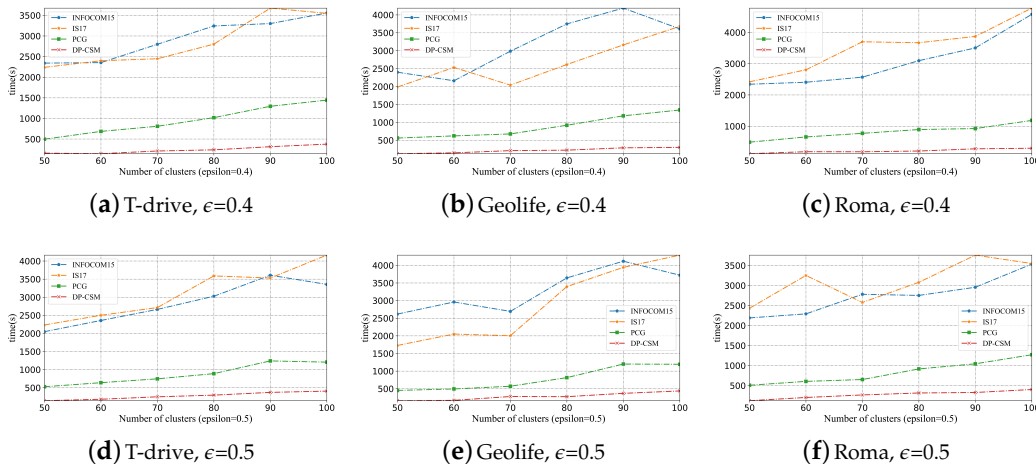

**Figure 12.** Impact of $k$ on the efficiency of the four methods. $k$ is a parameter of the $k$-means algorithm, representing the number of clusters. In these experiments, we fix the size as $|\mathcal{D}| = 2000$ for each datasets, and choose two kinds of privacy budget, i.e., $\epsilon = 0.4$, and $\epsilon = 0.5$. For each epsilon and dataset, we vary $k$ from 50 to 100, and compare the running times of the four methods.

## 6. Conclusions

In this paper, we present the DP-CSM, a differentially private trajectory synthesizing algorithm based on coresets and staircase mechanism for trajectory data publication. Compared with existing cluster center-based solutions, the DP-CSM utilizes the coreset to improve the efficiency and utilizes staircase mechanism to replace the traditional Laplace mechanism to improve utility. The DP-CSM is mainly composed of the following two steps: location generalization and trajectory reconstruction. In the first step, we construct the coreset of location set at each timestamp. Then, we use the *k*-means clustering to obtain the generalized location sets. In the second step, we first reconstruct trajectories and add noise to the count of reconstructed trajectories. According to the counts of reconstructed trajectories, we supplement trajectories and add noise. The experimental results show that the DP-CSM has greatly improved the efficiency while preserving a similar utility and privacy level to those of the three prevailing methods, such as INFOCOM15, IS17 and PCG.

In the future, we hope to eliminate the trajectory reconstruction step by directly constructing high-dimensional coresets based on the original trajectory, thus further improving the utility of the data and time efficiency. Additionally, DP-CSM can currently only process the trajectories with same length—we hope to overcome this insufficiency in the future.

**Author Contributions:** Conceptualization, Xin Yao; methodology, Xin Yao; software, Xin Yao; validation, Xin Yao; formal analysis, Xin Yao; investigation, Xin Yao; resources, Juan Yu; data curation, Xin Yao; writing—original draft preparation, Xin Yao; writing—review and editing, Xin Yao, Juan Yu, Jianmin Han, Yijia Wu, Xiaoqian Cao, Jianfeng Lu and Hao Peng; visualization, Xin Yao; supervision, Juan Yu and Jianmin Han; project administration, Juan Yu and Jianmin Han; funding acquisition, Juan Yu and Jianmin Han. All authors have read and agreed to the published version of the manuscript.

**Funding:** This research was funded by the National Natural Science Foundation of China under Grant No. 61702148 and Grant No. 61672648.

**Institutional Review Board Statement:** The study was conducted in accordance with the Declaration of Helsinki, and approved by the Institutional Review Board (or Ethics Committee) of Zhejiang Normal University.

**Informed Consent Statement:** Informed consent was obtained from all subjects involved in the study.

**Data Availability Statement:** Not applicable

**Conflicts of Interest:** The authors declare no conflict of interest.

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
