# Peer review of "DP-CSM: Efficient Differentially Private Synthesis for Human Mobility Trajectory with Coresets and Staircase Mechanism"

_ijgi, doi:10.3390/ijgi11120607_

Round 1

Reviewer 1 Report

The authors proposed a differentially private trajectory synthesizing algorithm called DP-CSM. This algorithm combines location generalization using k-means clustering and trajectory reconstruction by adding noise. They gave a theoretical proof that DP-CSM satisfies \epsilon-differential privacy. They also conducted experiments on three real-world trajectory datasets to evaluate the performance of the proposed method by comparing with existing methods. The paper is well organized. 

My comments are listed below:

1. The parameter \beta is not explained well. Please explain why you introduce \beta for the performance evaluation. 

2. Please explain how the differentially private property impacts on the resulting trajectory publication. For example, how can DP-CSM change in  the predictability of privacy-related places, such as locations of home and workplace compared to existing methods? 

3. In the performance evaluation, there are utility and scalability measures, but no privacy measure. How about comparing privacy measure by introducing the entropy of human mobility. 

Reference:  https://www.science.org/doi/10.1126/science.1177170

I would appreciate it if you comment on this. 

4. In Figure 8, there are no results when \epsilon=0.5 while there are in Figures 6 and 7. Is there any reason to omit the results? Also, please mention why you select \epsilon=0.4 and 0.5 for the performance evaluations.  

Reviewer 2 Report

Please, refer to the attached file.

Reviewer 3 Report

This paper proposed a very tool to protect the privacy of human mobility datasets. DP-CSM consisting of two steps are well described. Coreset-based clustering + staircase mechanism provided a promising idea to trade-off between utility and privacy. 

However, even though the paper has a very clear structure, the presentation remains to be improved. For example, lemma/theorem formalisation (should be more formal). 

More details are shown in the attachment. 

Round 2

Reviewer 3 Report

The authors addressed most of my concerns, here are some points that need to be considered:

1.     Figure 2 caption, it is better to demonstrate more explanations about circles and triangles. Also, the same problems for other figures and tables. The captions are usually too short to clarify the plots.

2.     Currently, I like your Theorem 2 because the assumptions and conclusions are very clear. However, in Theorem 1, you still used “denoting Algorithm 2 as XXXX”, what does mean? The result of algorithm 2? One particular step algorithm 2. The assumptions in the theorem should be specific. I strongly suggest you repeat some parts of Algorithm 2 and make it clear to put them in the theorem.

3.     Even though you added two measures based on entropy, however, I’m afraid that entropy can be used to measure the information or uncertainty rather than privacy. We have already had a privacy budget \epsilon, thus we don't need any other privacy measure. However, the reason why both reviewer 2 and I mention the predictability or entropy is to look at the utility. We have original data, we apply our algorithm, and we hope to get results that have similar utility but protect privacy. The trajectory similarity is just one of utility, as shown by plenty of human mobility research, predictability based on Lempel-Ziv entropy is also an important characteristic of human traces, so we hope to see if your algorithm can reserve some predictability patterns as a privacy-preserving method.

Maybe it is not compulsory for this article, but it should be very interesting and important in your future work.  
